# Predicting the Protective Behavioral Intentions for Parents with Young Children Living in Taipei City and New Taipei City Using the Theory of Planned Behavior for Air Polluted with PM2.5

**DOI:** 10.3390/ijerph20032518

**Published:** 2023-01-31

**Authors:** Siu-Kei Woo, Chih-Jui Pai, Yi-Te Chiang, Wei-Ta Fang

**Affiliations:** Graduate Institute of Environmental Education, National Taiwan Normal University, Taipei 116, Taiwan

**Keywords:** PM2.5, environmental behavior, behavioral intentions, theory of planned behavior, path analysis

## Abstract

While studies on the damaging effects of PM2.5 air pollution are abundant, studies seeking to understand the factors that influence human behaviors for the avoidance of exposure to PM2.5 are lacking. Theory of Planned Behavior (TPB) can be used to investigate the effects of Attitudes (AT), Subjective Norms (SN), and Perceived Behavioral Controls (PBC) in the Behavioral Intentions (BI) of parents with young children against exposure to PM2.5. Questionnaires, based on the TPB used to predict BI, were distributed to 610 parents in Taipei City and New Taipei City. Our results revealed that the AT of both groups had a significant positive predictive effect on their PBC and BI. While the SN of the Taipei group affected BI directly, there was no significant effect on the BI from the SN of the New Taipei group. Using path analysis, Taipei City and New Taipei City groups had different BI paths: All five hypotheses are statistically significant and form four paths in the Taipei City group. While only four hypotheses in the New Taipei City group formed three paths and no path for SN-BI. Both groups formed behaviors that were based on the SN/PBC around them, which ultimately contributed to the BI of their protective behaviors.

## 1. Introduction

Studies linking awareness of PM2.5 pollution to personal behaviors seeking to avoid such pollution are scant, and there is a lack of focus on this issue in Asia. The East Asia region, due to its economic development, rapid industrialization, and urbanization, has some of the highest levels of air pollution in the world, yet few subnational risk assessment studies on air pollution have been conducted in Asian countries [1]. Shi et al. [2] found that few studies looked at behaviors aimed at reducing exposure to PM2.5 in urban among urban residents. In addressing severe haze pollution, it was found that a household’s environmental concern, indirectly and significantly affects occupant’s moral norms and intentions [2].

This study will attempt to understand the factors influencing anti-PM2.5 behavioral intentions, by adapting the theory of planned behavior framework to investigate the effects of attitudes, subjective norms, and perceived behavioral control on protective behavioral intentions against PM2.5 [3]. It will explore the effects of attitudes, subjective norms, and perceived behavioral controls on the behavioral intentions of parents with young children. Predicting the behavioral intentions of parents is essential because of the impact of air pollution on individual behavioral intentions and the issue of air pollution is closely related to sustainable development, and to society at large. The application of the Theory of Planned Behavior, in the field of environmental pollution and air pollution, is important [3,4,5]. Air pollution affects human behavior at the micro level, and previous research has investigated the public health risks associated with atmospheric exposure to PM2.5 for different subpopulations, including race and age [5]. There is a lack of research into variations in parental protection behavior at the family level in Asia [1]. Through this research, we hope to learn the relationships between behavior and environmental issues, through understanding parents’ prevention behavior and how this plays out through their children who are exposed to air pollution with PM2.5. In doing so, we also seek a greater understanding of the relationship between psychosocial factors and anti-PM2.5 behavioral intentions [6].

There appears to be general agreement among social psychologists that most human behavior is goal-directed [7,8]. According to Ajzen [9], being neither capricious nor frivolous, human social behavior can best be described as more or less well-formulated plans. Actions are controlled by intentions, but not all intentions are carried out; some are abandoned altogether while others are revised to fit changing circumstances [10].

There is a lack of research into this topic, and this is made more glaring considering the increasing impetus globally to deal with the problem of air pollution. Of the United Nations Sustainable Development Goals (SDGs) [11], air pollution is directly or indirectly related to all 17 SDGs to varying degrees [12] with 169 items within the 17 goals marked as directly related to air pollution and 8 targets explicitly aimed at reducing it. Elder and Zusman [13] have highlighted air pollution’s fundamental importance to the SDGs and its impact on the diverse issues involved, including affordable clean energy, dignified employment, economic development, industrial innovation and infrastructure, responsible consumption and production, and transport in sustainable cities and communities. Air pollution is also recognized as an important contributing factor to disease globally [12]. Lode et al. [14] proclaim the SDGs as an opportunity to fight air pollution globally. Thus, the global promotion of the SDGs and their aimed realization by 2030 are intricately tied to problems resulting from air pollution and are crucial for its reduction [15]. 

In Taiwan, air pollution—especially PM2.5 pollution—is a major environmental issue. Although the level of pollution varies greatly in different areas of Taiwan, significant levels do occur in Taipei City and New Taipei City. One reason for such high incidences in these two cities is the high population and population densities there. New Taipei City has the highest number of deaths attributable to PM2.5 in Taiwan at 874 people/year, while Taipei City is relatively lower, but still high at 619 people/year [16].

According to Lo et al. [16], the risk assessment of PM 2.5 is important, not only at the global and national levels, but also at the local, county, and city levels, due to its high variability between locations. Understanding the impact of PM2.5 exposure locally can assist central and local governments in the formulation and development of public health and environmental policies. The relationship between local-level analysis, and air pollution prevention and control behavior has been shown to be important [17], therefore a deeper understanding of local influences and behaviors between Taipei City and New Taipei City is significant, not only for the local and national understanding of the issues and potential solutions available but also for our understanding globally of the factors involved in behaviors and attitudes for the prevention of damage caused by PM2.5 air pollution.

## 2. Study Area

In this study about PM2.5 air pollution in Taipei City and New Taipei City, living location is a key factor for our investigation. There are differences in the level of urbanization between Taipei City and New Taipei City and this influences the categorization of our study areas. As a result, in accordance with ‘Lo Chi-hon Stratification’ [18] and Liu et al. [19], the six districts of Sanchong, Zhonghe, Yonghe, Banqiao, Xindian, and Xinzhuang have been included in our Greater Taipei group rather than the New Taipei City Group-due to the former’s higher urbanization level. In this study, ‘New Taipei City Group’ thus refers to New Taipei City minus the six townships mentioned above, and ‘Greater Taipei Metropolitan Group’ refers to Taipei City and those additional six townships (Figure 1).

According to the Department of Budget, Accounting and Statistics, Taipei City Government [20], the population of Taipei City is 2,524,393 with a population density of 9288/km^2^ in 2021. According to the Department of Budget, Accounting, and Statistics, New Taipei City Government [20], the population of New Taipei City is 4,030,954 and the population density is 1964/km^2^ in 2021. In this research, the six townships in New Taipei City with the highest population were grouped into the Greater Taipei Metropolitan Group (Figure 1) due to their population and population density being more similar to that of Taipei City than that of New Taipei City. The six townships and their population and population density are as follows: Banqiao (557,114 and 24,079/km^2^), Sanchong (385,328 and 23,615/km^2^), Zhonghe (411,214 and 20,414/km^2^), Yonghe (219,364 and 38,392/km^2^), Xinzhuang (422,653 and 21,413/km^2^), and Xindian (303,532 and 2525/km^2^). These six townships exhibit an urbanization level similar to Taipei City.

There are numerous definitions of ‘ex-urban’, and the spatial boundaries of such labeled areas are often uncertain [21]. According to ‘Lo Chi-hon Stratification’ [18], seven factors are used to identify the spatial boundaries of ex-urban areas mentioned by Ban and Ahlqvist [21], including population, population density, demographic characteristics, industrial development, public facilities, financial situation, and geographical environment [18].

When considering the urban geography of Taiwan, ‘Lo Chi-hon Stratification’ [18] is an often-cited city hierarchy model that is used to determine geographical and socio-economic divisions in Taipei City and New Taipei City. ‘Luo Chi-hon Stratification’ has been widely used in large-scale surveys involving hierarchical sampling in townships and urban areas of Taiwan, such as the “Basic Survey Plan on Social Change in Taiwan”, by the Academia Sinica, and by Hsiao [22] in conducting a study of “Environmental Awareness and Value in Taiwan”. Since our study focuses on air pollution control behavior, demographic characteristics including population and population density are vitally important, and will help explain how the demographic characteristics of Taipei City and New Taipei City affect the hypothesis that residents in these two regions have different air pollution control behaviors.

For hierarchical regional geographic models of cities, the Central Place Theory by Christaller [23] can be used to analyze relationships between towns and cities in the same area. However, the Central Place theory has been criticized as too static, as it does not incorporate the element of time into the development of the central place. In addition, whilst the theory works well in the agricultural sector, it does not satisfactorily apply to industrial or post-industrial areas, because of the diversification of their various services and the uneven distribution of natural resources within [24]. For this reason, this study uses instead the ‘Urban Dense Zone Model’ (Desakota) [25], which is the opposite of ‘Megalopolis’ as described by Gottman [26]. ‘Desakota’ is instead described as a ‘City-village Process’, in an Extended Metropolis Region (EMR) and derives from the ‘Theory of Core-periphery’ proposed by the American-regional planning scholar Friedmann [22], which states that any region has a ‘Core’, together with multiple ‘Peripheral Regions’. In this model, the core area has a decisive influence on the periphery, while there is simultaneously a high degree of economic growth in the overall metropolitan area.

Fann et al. in the United States found that air pollution in different ‘Regional Spaces’ causes different health hazards [27] and there are large county-to-city differences in PM2.5 in Taiwan. Concentration levels are closely related to the degree of urbanization, industrial development, and of pollutant transmission [16]. Everard and Longhurst [15] argue that clean air is a common human asset and that breathing clean air is a universal human right. However, countries with different levels of development have encountered problems in controlling air quality. Underdeveloped countries actually produce relatively low volumes of air pollution, but they necessarily share cross-regional risks and hazards for the air pollution generated by the economic and industrial development in more highly developed countries. Therefore, air pollution is not a problem for any single individual or country, but a globally important issue that affects the common life and health of all human beings, according to the World Health Organization (WHO).

## 3. Air Pollution Situation in Taiwan and the World

### 3.1. Air Pollution Situation in the World

Research over the past few decades has shown that exposure to fine suspended particulates (PM2.5) poses a significant health hazard. Large epidemiological studies have found a significant association between PM2.5 and death from cardiopulmonary disease and lung cancer [28,29,30,31]. In addition, studies in the United States have found that reducing exposure to PM2.5 may increase average life expectancy [32]. In 2004, the American Heart Association published a study on air pollution and cardiovascular disease, showing that exposure to PM2.5 air pollution is associated with a significantly increased risk of cardiovascular-related diseases and even mortality [33].

The WHO describes air pollution as the world’s greatest environmental health threat, estimating that it causes 4.2 million premature deaths each year. Many cancers are also directly or indirectly linked to air pollution, and lung cancer is the most commonly diagnosed cancer worldwide, with an estimated 2.1 million new cases and 1.8 million deaths worldwide in 2018—accounting for 11.6% of all newly diagnosed cancers and 18.4% of cancer deaths [34,35]. Due to the variability of air pollution in spatial areas, the risk assessment of PM2.5 is important not only at the global and national levels, but also at the local county and city levels. Understanding the impact of PM2.5 exposure at the national and local county and city levels can assist central and local governments in the formulation and development of public health and environmental policies [18].

According to the Global Urban Ambient Air Pollution Database, more than 80% of people living in urban areas do so with air pollution that exceeds WHO standards [36]. The study further shows that when air pollution increases, so does the number of deaths, and it is estimated that outdoor air pollution will cause a rapid increase in premature mortality by 2050. However, raising awareness of air pollution is an important step in reducing air pollution and improving people’s health [37,38]. Many large cities around the world are facing serious environmental pollution challenges, including high air pollution emissions [39,40]. Large cities tend to be high-risk areas, and urban dwellers are vulnerable to health hazards caused by air pollution, so this risk needs to be estimated to improve the sustainability of life in large cities worldwide [41].

The WHO published data and information on air pollution in 2018, noting that air pollution is a major environmental risk affecting health. Air pollution is a major environmental health problem for nationals of developed and developing countries, with 430 thousand people dying from diseases caused by indoor air pollution per year, and 3.7 million people dying from outdoor (ambient) air pollution about 88% of these premature deaths occur in low-and middle-income countries. Environmental air pollution, chemical pollution, and soil pollution, as well as pollution from industry, mining, power generation, mechanized agriculture, and petroleum-powered vehicles, have also been found to be increasingly high, most notably in low and middle-income countries [42].

### 3.2. The Air Pollution Situation in Taiwan

Of the top ten causes of death in Taiwan in 2017, announced in 2020 by the Ministry of Health and Welfare, most are directly or indirectly related to air pollution, and the harm caused by air pollution cannot therefore be underestimated by the Ministry of Health and Welfare in 2020. Air quality testing is based on the estimation of pollutants in the air, such as the main gaseous pollutants: nitrogen dioxide (NO_2_), sulfur dioxide (SO_2_), nitric oxide (NO), ozone (O_3_), particulate pollutants (Fine Particulate Matter ≤ 2.5 μm, PM2.5), and PM10 [43].

In order to protect air quality, Taiwan enacted the Air Pollution Prevention and Control Act in 1975 to standardize air quality standards, and gradually established various monitoring stations throughout Taiwan, with 78 stations to monitor air quality and provide early warning. However, because air pollution has the characteristics of asymmetric distribution of derivation, diffusion, and accumulation, the real source of pollution emission is not necessarily located in affected areas; in addition, different climatic conditions often affect the degree of pollution. It is precisely because of the complexity and difficulty of air pollution control and monitoring that it presents such a risk to society [44].

The average annual PM2.5 exposure concentration at the county and city level in Taiwan is taken from the Taiwan Air Quality Monitoring Network [45]. When there are several monitoring stations in a single county or city, the figures from township areas with population densities greater than 10,000 (people/km^2^) (areas with the highest population density) are used. Monitoring station values are then representative of the majority of the population in a given county or city [45].

In recent years, the concentration of PM2.5 in Taiwan has been declining, and the average PM2.5 concentration in Taiwan has been 36.2 μg/m^3^ since 2005. This dropped to 25.0 μg/m^3^ in 2014, but today’s PM2.5 concentration is still far higher than that recommended by the World Health Organization [46,47]. People living in the same area are often exposed to PM2.5 pollution comparable to the health risk factors of individual lifestyle choices, such as smoking, drinking, and not exercising, so the effect of avoiding PM2.5 exposure through personal action is often limited. This highlights the vital role of governments in addressing environmental health risks [18].

## 4. Theories and Hypotheses

In the Rational Choice Theory by Ajzen (1991), he utilizes TPB. The TPB is a commonly used research model used to predict BI [6,48] and is one of the most commonly used theories in research studies on the prediction of environmental behaviors [49,50]. To study behaviors for avoiding air polluted with PM2.5, attitude, perceived behavioral controls, and subjective norms are known to shape such protective behaviors. The TPB model focuses on the controlled aspects of human decision making and information processing of behaviors, which are guided by a conscious self-regulatory process and by goal-oriented behaviors. One’s AT can affect BI indirectly or directly through one’s SN [51].

Many environmental education studies have found that environmental action cannot only rely on the improvement of knowledge and technology, but also requires the promotion of psychological traits such as affection and attitude [52]. One example is the recently released “Taiwan Parents’ Air Pollution Awareness and Children’s Impact Survey Report” [53], which found that 91.4% of parents are aware that air pollution is becoming worse, but that 84.2% of parents have not established the most basic habits for checking air quality. It can therefore be seen that the improvement of awareness of air pollution, and of technology to monitor it, is not enough to promote the occurrence of actions, an individual’s assessment of air pollution risks, and their willingness to take action also influences actions to prevent and control air pollution [12].

According to the TPB, attitude refers to the degree of a person’s support for a particular behavior [10]. The definition of subjective norms is that an individual perceives the overall behavioral perception that a group expects to perform [10], perceived behavioral controls are defined as the difficulty in consciously performing a particular behavior [10], and behavioral intentions refer to the degree to which people are inclined to engage in a particular behavior [10]. Personal factor is an individual’s positive or negative evaluation of performing a behavior, this factor is also termed attitude [10]. The second determinant of intention is a person’s perception of the social pressures put on them to perform, or not perform, a given behavior. Since it deals with perceived prescriptions, this factor is termed subjective norm. Generally speaking, people intend to perform a behavior when they evaluate it positively and when they believe that people whom they deem as important think they should perform it [10].

Adding to the empirical evidence from previous studies in relation to the study hypotheses, the use of this model may help explain the relationship between social psychology and people’s preventative behaviors related to environmental problems. Despite the many studies assessing the toxicity and impact of air polluted with PM2.5 on humans and the environment, work on the factors that influence people’s behaviors such as AT and SN toward avoiding air polluted with PM2.5 is lacking [46]. From the perspective of TPB, it has been found that even when accounting for the predictive variables in TPB, understanding past behaviors can help predict future behaviors [6].

Insight into the factors that affect people’s behaviors, such as AT and SN, for avoiding air pollution is lacking [54]. In Figure 2, we tested the following hypotheses.

**Hypothesis 1a and 1b (H1a, H1b).** *The Greater Taipei Metropolitan Group (H1a) and New Taipei City Group (H1b) possess AT that can affect their respective BI [51,55,56,57,58,59]*.

**Hypothesis 2a and 2b (H2a, H2b).** *The Greater Taipei Metropolitan Group (H2a) and New Taipei City Group (H2b) possess PBC that can affect their respective BI. The PBC of parents of both groups affects their BI directly [56,57,60]*.

**Hypothesis 3a and 3b (H3a, H3b).** *AT can affect the PBC of the Greater Taipei Metropolitan Group (H3a) and New Taipei City Group (H3b). The results of studies indicate people’s SN and PBC are significantly related [61,62,63]*.

**Hypothesis 4a and 4b (H4a, H4b).** *The Greater Taipei Metropolitan Group (H4a) and New Taipei City Group (H4b) possess SN that can affect their respective BI. Their SN and PBC affect BI [64,65], we hypothesized that the PBC of parents with young children living in Taipei and New Taipei affect their SN positively. The results of studies have shown that people’s SN and PBC are significantly related [61,62,63]*.

**Hypothesis 5a and 5b (H5a, H5b).** *The Greater Taipei Metropolitan Group (H5a) and New Taipei City Group (H5b) follow SN that can affect their respective PBC. Their SN affects their PBC directly and affects BI indirectly [60]*.

## 5. Methodology

According to Ajzen [10], numerous studies supporting the theory of rational action have been advanced through various experiments. In order to provide a comprehensive test of the relationship indicated by the theory, it is necessary to present behavioral intention through pilot studies and then use these theories to construct a standard questionnaire. The resultant questionnaire will contain measurements of the following variables:(1)Behavioral assessments, which are assumed to determine attitudes toward behavior;(2)The motivation to comply with subjective norms;(3)Direct measurement of attitudes and subjective norms;(4)The behavioral intention to perform the act [10].

Our study followed a purposive and stratified methodology [66] utilizing Liu’s 2018 approach [3] to develop, deliver, and collect questionnaires. The participating parents were contacted online via the ‘SurveyCake’ platform. The study received 610 questionnaires using an anonymous answering method online. The questionnaire was adapted from previous environmental behavior surveys, especially those involving AT [67], SN, PBC [68], and BI [67]. The questionnaire was reviewed by six experts of environmental education and health promotion.

A five-point Likert scale was used, as well as statistical software (Smart PLS 3.0, SmartPLS GmbH, Oststeinbek, Germany) for statistical and path analyses. Such analyses are critically important for studies such as ours involving small samples of populations, and serious health concerns in vulnerable populations [69].

For the demographic characteristics of the 610 respondents, there are 180 males and 303 females in the Greater Taipei Metropolitan group, and 56 males and 71 females in the New Taipei City group. The educational attainment of the parents involved is mainly at the university graduate level, with 62.3% and 59.8% in the Greater Taipei Metropolitan Area and New Taipei City, respectively. This is followed by high school graduate level, with 26.3% in the Greater Taipei Metropolitan area, and 33.8% in New Taipei City.

Regarding the entire yearly income of the participating families, 28% and 29.9% of parents earned more than one million Taiwan dollars in the Greater Taipei Metropolitan area and the New Taipei City, respectively; 15.7% of families in the Greater Taipei Metropolitan earned 0.9 to 1 million Taiwan dollars annually; and 18.1% of families in New Taipei City earned 0.7 to 0.8 Taiwan dollars. Regarding the parents’ occupation as a percentage of the total, the service industry was highest (35% and 34.6% in the Greater Taipei Metropolitan area and New Taipei City, respectively), then manufacturing (28.2% and 26.8%), and then housewife (14.5% and 14.2%). The interpretations are based on sound science methodology and the study was data led [10].

## 6. Results

In Table 1, Table 2, Table 3 and Table 4, the Cronbach α for questions posed to the parents from both groups supports the reliability of the questions [69]. The Cronbach α value of the questionnaire of the study was 0.948 and exceeds 0.7, which is considered good.

### 6.1. Correlation Analysis

The path analysis using PLS-SEM was used to verify the hypotheses of the research [70]. A correlation analysis was performed on the average score results of the parents in Table 5 and Table 6.

### 6.2. Path Analysis and PLS-SEM

Aspects of the BI influencing the parents of both groups were analyzed using SEM software (Smart PLS 3.0, SmartPLS GmbH, Oststeinbek, Germany) and the results are shown in Table 7 and Table 8. The AVE is a measure of the amount of variance captured by a construct in relation to the mount of variance due to error [26]. The values of both groups are greater than 0.5, which is the acceptable value factor loadings [71]. The CR is a measure of the internal consistency of a scale item [72]. The Cronbach’s α of the four dimensions all reached a credibility standard of more than 0.5 [73,74], indicating that all of the data the study collected are valid. The t-value of the paths were obtained using BootStrapping methodology to test the significance levels of the results [75].

In the Greater Taipei Metropolitan Group, the AT had a significant positive predictive effect on their PBC (β = 0.447, t = 10.252 ***) and BI (β = 0.230, t = 4.516 ***). The SN had a significant positive predictive effect on their PBC (β = 0.445, t = 10.155 ***) and also BI (β = 0.315, t = 6.201 ***). The PBC had a significant positive predictive effect on their BI (β = 0.334, t = 5.547 ***) (see Figure 3).

In the New Taipei City Group, the AT had a significant positive predictive effect on their PBC (β = 0.285, t = 3.368 ***) and BI (β = 0.295, t = 2.870 **). The SN had a significant positive predictive effect on their PBC (β = 0.632, t = 7.384 ***) but no significant effect on their BI (β = 0.192, t = 1.669). The PBC had a significant positive predictive effect on their BI (β = 0.387, t = 2.967 **) (see Figure 4).

## 7. Discussions and Implications

According to Gifford and Nilsson [76], cultural and ethnic differences can affect pro-environmental behaviors. Fang [48] pointed out that “There are often different environmental concerns between different races and ethnic groups, and cognitive differences can emerge due to cultural differences”. Fang [48] also suggested that these differences were related to overall thought structures and the logic of different cultures. As with Taiwan, the significance of living location for influencing outlook and norms is important in other Asian cultures [77]. There is a lack of research on parental protection behavior in Taiwan in relation to different living locations. Therefore, this study is significant and contributes to a theoretical and practical understanding of this issue. Some studies have argued that children, through their learning about the environment from school education, can also transmit environmental knowledge and attitudes to their parents, thus affecting their parents’ environmental behaviors [78,79].

Table 9 compares the results of parents living in different locations. In the New Taipei City Group, there is no statistical significance in the SN to BI. Meanwhile, there is statistical significance in the SN to BI of the Greater Taipei Metropolitan Group. The decision-making process of the parents living in Greater Taipei Metropolitan was more dependent on opinions from others and this lead to the statistical significance in the SN to BI, which indicates those parents’ responsiveness to opinions of people close to them [80]. The higher level of education in Greater Taipei Metropolitan Group might also influence personal attitudes, which could further affect subjective norms [81].

Both the Greater Taipei Metropolitan and the New Taipei City Group believed they could limit exposure to air pollution with PM2.5, so there is statistical significance in their PBC to BI. The PBC of the Greater Taipei Metropolitan Group was also affected by AT and SN, which suggests that their PBC is influenced by SN and the people around them, but that they can also develop their own AT toward air pollution protection behaviors [61,62,63]. This could be related to higher levels of education and income among the Greater Taipei Metropolitan Group, which has a strong correlation with PBC [2].

### 7.1. Influence of Attitude (AT)

Attitude refers to the degree of a person’s support for a particular behavior according to the TPB [6]. Our study shows that there are differences in the AT of the two groups with respect to protective behaviors related to air pollution with PM2.5. The path model indicates the AT of both groups can affect their BI directly (β = 0.230, t = 4.516 > 3.29 for the Greater Taipei Metropolitan Group and β = 0.295, t = 2.870 > 2.58 for the New Taipei City Group). These results were demonstrated in the four paths of the Greater Taipei Metropolitan Group and three paths of the New Taipei City Group. The study also shows that there is no direct path for AT-BI in the New Taipei City Group.

The attitude of parents living in New Taipei City, which is more rural compared with the Greater Taipei Metropolitan Group, in regard to their anti-PM2.5 behavior intentions aligned with other empirical studies [2,60,82,83,84]. In rural societies, positive attitude is the most important factor in influencing behavioral intentions, followed by internality PBC and subjective norms [81].

### 7.2. Influence of Subjective Norm (SN)

The definition of Subjective Norms is that an individual perceives the overall behavior that a group expects to perform [10]. This study looked at the extent to which an individual’s interaction with others, such as family and friends, will affect one’s views and positions on air pollution, as well as in response to the perceived pressure due to public opinion, news media, and private environmental groups.

These results show that the SN of both groups can directly affect their PBC (Greater Taipei Metropolitan Group: β = 0.445, t = 10.155 > 3.29, New Taipei City Group β = 0.632, t = 7.384 > 3.29); however, only the SN of the Greater Taipei Metropolitan Group can affect BI (Greater Taipei Metropolitan Group: β = 0.315, t = 6.201 > 3.29, New Taipei City Group β = 0.192, t = 1.669 < 1.96). This suggests that the BI of both Groups with regard to air pollution is based on the expectations of people around them, especially in the Greater Taipei Metropolitan Group. SN is influenced by the expectations of those perceived as important to the individual [6].

This is particularly relevant in the context of Asian society, where norms are more greatly valued [3]. Individuals are usually willing to fulfill the expectations of other individuals or organizations, potentially due to collectivism as a longstanding characteristic of traditional culture. Asian residents are more concerned about the views of other individuals or organizations when deciding whether to take certain actions [2].

### 7.3. Influence of Perceived Behavioral Control (PBC)

Ajzen [85] added the Perceived Behavioral Controls variable (anticipating the likelihood of success or failure of what one wanted to do in the future) to the Theory of Planned Behavior, in order to provide a better interpretation and prediction of behavior, thus forming the framework of planned behavior theory. It is often used to predict behavioral intentions that have positive meaning for individuals or society.

Our research shows that there is statistical significance in the PCB of air pollution prevention behaviors in both groups, greater than SN (Greater Taipei Metropolitan Group: β = 0.334, t = 5.547 > 3.29; New Taipei City Group β = 0.387, t = 2.967 > 2.58). The PBC for the parents of both groups affects their respective BI [2,56,72,86] associated with air pollution prevention, and that PBC is the main factor influencing their BI.

The PBC of both groups is affected by the same factors in our study. The PBC of both groups was affected by their SN, which has been demonstrated in other research [46,47,48]. The results suggest that the New Taipei City Group possesses a PBC that is affected by SN, rather than AT toward air pollution protection behaviors intention (BI), but that AT can positively affect PBC directly [62,86].

### 7.4. Analysis of Behavioral Intention (BI)

Behavioral Intentions refer to the degree to which people are inclined to engage in a particular behavior [9]. The behavior is generated by logical thinking of Behavioral Intentions: after the selection, if other decisions contribute to the behavior, then the higher the people’s environmental Behavioral Intentions, and so the higher the actual execution rate of the behavior. In this study, one of the objectives is to find out the degree of willingness to understand the problem of air pollution, cooperate with government policies, and adopt decisive strategies.

The results of the *t*-test show that the air pollution prevention behavior of the Greater Taipei Metropolitan Group was statistically lower than that of the New Taipei City Group, which means the New Taipei City Group has greater prevention behavior than the Greater Taipei Metropolitan Group toward air polluted with PM2.5.

Only four of the five hypotheses in the New Taipei City Group were supported and formed three paths including AT to BI, AT to PBC to BI, and SN to PBC to BI. The PBC in the New Taipei City Group was based on the SN and PBC around them, which ultimately contributed to the BI of their protective behavior. In the Greater Taipei Metropolitan Group model, the five hypotheses were all supported. Statistically higher scores showed a more diverse and complex model and formed four paths including AT to BI, AT to PBC to BI, SN to BI, and SN to PBC to BI.

In the New Taipei City Group model, four hypotheses were supported. Only the SN to BI path was not supported. The BI generated in the Greater Taipei Metropolitan Group shows a series of more complex and diverse pathways that have significantly lower scores than the New Taipei City Group, which means the decision progress of the Greater Taipei Metropolitan Group is more complex than those of the New Taipei City Group. The Greater Taipei Metropolitan Group can form BI from SN directly. Previous research suggests that higher education level may contribute to the above situation [26].

The study found a difference between SN and BI. The SN of the New Taipei City Group does not directly affect their BI, but the SN is affected indirectly through the BI [60] in the Taipei City Group. This means that those parents’ expectations toward the opinions of people whom they care about the most [6]. The higher level of education in the Greater Taipei Metropolitan Group might also influence personal attitudes, which could further affect subjective norms [77,87]. Besides the direct influence of the BI, the New Taipei City Group are influenced by their SN [64,65]. This means the BI in both groups is partly derived from meeting the expectations of others around them, not just their own PBC, but via pressure exerted by SN and cultural values.

### 7.5. t-Test

The result of the independent sample *t*-test in Table 10 found that the AT, SN, PBC, and BI of the New Taipei City Group are statistically higher than the Greater Taipei Metropolitan Group. This means that parents in the New Taipei City Group agree more with the questionnaire statements.

However, SN does not affect BI in the New Taipei City Group and even the scores in the *t*-test are higher than in the Greater Taipei Metropolitan Group. This may be as a result of both SN and PBC being high in the *t*-test, thus decreasing the importance of SN in terms of affecting BI.

## 8. Conclusions, Limitations, and Future Research

### 8.1. Conclusions

While studies looking at the effects of air pollution and PM2.5 are abundant, research into understanding behaviors for avoiding exposure to PM2.5 in Taiwan is lacking. Questionnaires based on the TPB used to predict BI were distributed to 610 parents in Taipei and New Taipei. Our results revealed that the AT of both groups had a significant positive predictive effect on their PBC and BI. While the SN of the Taipei group affected BI directly, there was no significant effect on the BI due to the SN of the New Taipei group. Using path analysis, it was apparent that the Taipei and New Taipei groups had different BI paths. All five hypotheses are statistically significant in four paths in the Taipei group. While only four hypotheses in the New Taipei group formed three paths and no path for SN-BI. Both groups formed behaviors that were based on the SN/PBC around them, and these ultimately contributed to the BI of their protective behaviors.

Path analysis demonstrated that the Greater Taipei Metropolitan Group and New Taipei City Group had paths with different BI. There are more paths of BI in the Greater Taipei Metropolitan Group than in the New Taipei City Group: In the five hypotheses, four of them forming paths including AT to BI, AT to PBC to BI, and SN to PBC to BI in the New Taipei City Group. In the New Taipei City Group, the PBC was based on the SN around them, which ultimately contributed to their protective behavior and BI. There is no path for SN to BI in the New Taipei City Group. In the Greater Taipei Metropolitan Group, the five hypotheses including H1a, 2a, 3a, 4a, and 5a provided statistically high scores, which form paths AT to BI, AT to PBC to BI, SN to PBC to BI, and SN to BI. We thus can see the differences in paths between the two groups in this study.

According to Gifford and Nilsson [76], personal and social factors can affect pro-environmental behavior. It is worth performing more research that considers the types of factors that affect environmental behaviors toward air pollution. They [76] also pointed out that pro-environmental behaviors may be caused by personal non-environmental goals such as improving health and saving money. This aligns with other empirical studies with similar findings [57,82].

The findings of this study provide further insights into regulatory and health education for urban and rural parents regarding protection against PM2.5. According to Liu et al. [3], the government should continue promoting protection against serious smog pollution and PM2.5 through various channels, such as school education, community mass media, public service advertisements, and celebrity endorsements. In addition, the government should also encourage the strengthening of school environmental protection and health education, and the use of home–school cooperation, especially for the education of young children’s parents, so as to improve PM2.5 protection in terms of PBC [3].

### 8.2. Limitations

Economics, age, and sex are factors that affect prevention behavior. Liu et al. [3] and Woo [4] determined education as a key factor. In that study, behavior intentions were found between parents with postgraduate-level education and those with undergraduate education only. Parents with an undergraduate level education exhibited no significant route from AT to PBC, and also no significant route from SN to BI, while the parents with postgraduate education demonstrated a statistically significant route from AT to PBC and SN to BI. However, both parents with postgraduate and those with undergraduate attainment showed no significant route from AT to BI [4].

Our study only looked at the differences between people living in New Taipei City and those in the Greater Taipei Metropolitan area. We do not know how the education level of the parents involved influenced the results. Studies have shown that more educated people are more likely to recognize the negative health effects of air pollution, and these individuals are more likely to express concerns about air pollution, meaning they are more sensitive to the effects of air pollution [88].

In addition, Taipei and New Taipei City both have large metropolitan populations in both urban and rural settings. However, the findings cannot be generalized to other cities with large populations [3]. Some studies have found that people living near more natural environments are healthier than those who do not, and that the long-term indirect effects include family satisfaction and improved work and living standards [89]. New Taipei City has a more natural environment and more public spaces than Taipei City, this factor may also have contributed to the results of this research.

### 8.3. Further Studies

In Taiwan and in many Asian cultures, the significance of living location is important in determining one’s values and outlooks, and this is much more pronounced than in the Western world [77]. In Asia, living location is viewed as important to one’s identity and social status and it would be worthwhile to compare how such influences determine air pollution prevention behaviors in other cultures.

According to Liu et al. [3], a more representative sampling of populations should be sought and tested to provide context to the findings. Therefore, studies from other countries should be gained for international comparisons and to ascertain similarities or differences. In addition, future studies may seek to explore the impacts of protection against PM2.5 in connection with environmental sensitivity, social norms, and other factors related to behavior [3].

The findings of this study provide insights into the health risk factors of individual lifestyles. People living in the same area are often exposed to similar PM2.5 pollution, so efforts to avoid PM2.5 exposure through individual behaviors are often very effective, highlighting an important role for governments in dealing with health risk factors in the environment [16]. Studies show that women, the elderly, and people in poor health are more concerned about the risk of air pollution [16] and such outlooks require further attention and elaboration.

As public awareness about air pollution increases, populations are realizing the damaging effects of unclean air on the body. An understanding of the negative social and economic outcomes may lead to greater pressure on governments to act [12]. Therefore, policy on air pollution and PM2.5 prevention also requires further insight.

Ongoing research into students in Taiwan and their support for policy dealing with air pollution prevention [12] is important. Students will have great influence on the future decisions and direction of governments and thus the future state of the environment.

Taiwan’s government has provided funds to improve power generation facilities, as well as the replacement of phase I and II diesel trucks, two-stroke engines, and boilers. They actively audit construction projects, ban open combustion, and regulate exhaust inspections of automobiles [90]. Further study can cooperate with the government’s Environmental Protection Administration, and the findings applied to real-world policy and education.

Taiwan’s current air pollution policy includes the adoption of air pollution control since 2017, the implementation of an air pollution control action plan, and the amendment of air pollution control laws leading to a more complete air quality management system [90]. Environmental protection policies are often mandatory, including regulations to limit emissions, the use of required technologies, and carbon taxes that charge emitters. Other measures include education programs and subsidies for acquiring low-carbon technologies [91]. All of these acts are beneficial to society and the environment but need huge amounts of research to connect them to individual behaviors and support for environmental policy in living environmental education [92].

For parents living in New Taipei City, a less urbanized area than the Greater Taipei Metropolitan area, the government should establish and improve policies such as the promotion of protective masks [3]. For urban parents living in the Greater Taipei Metropolitan Area, who have higher educational levels and PBC advantages, the government should encourage the acquirement of PM2.5 emission reduction and health protection products [3].

Finally, some studies of attitudes and perceptions of PM2.5 air pollution involve college and university students [12], but few studies have looked at younger children and the air pollution prevention behaviors of their parents. Parents are an essential factor in determining children’s environmental prevention behavior and behavior intention, especially in East Asia cultures where children are expected to obey the wishes of parents and grandparents. There is, therefore, scope for research on prevention behavior in relation to air pollution with PM2.5 amongst parents and children of different backgrounds and cultures, in countries all over the world.

## Figures and Tables

**Figure 1 ijerph-20-02518-f001:**
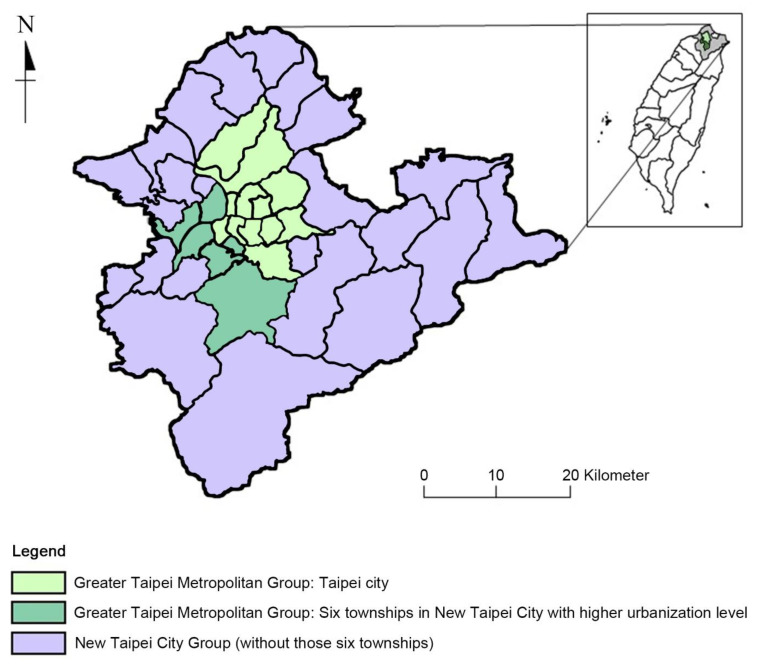
Study area ‘Greater Taipei Metropolitan Group’ (green) including Taipei City (light green) and six townships with a higher urbanization level (deep green): Sanchong, Zhonghe, Yonghe, Banqiao, Xindian, and Xinzhuang; ‘New Taipei City Group’ including New Taipei City without those six townships (purple) [15,16].

**Figure 2 ijerph-20-02518-f002:**
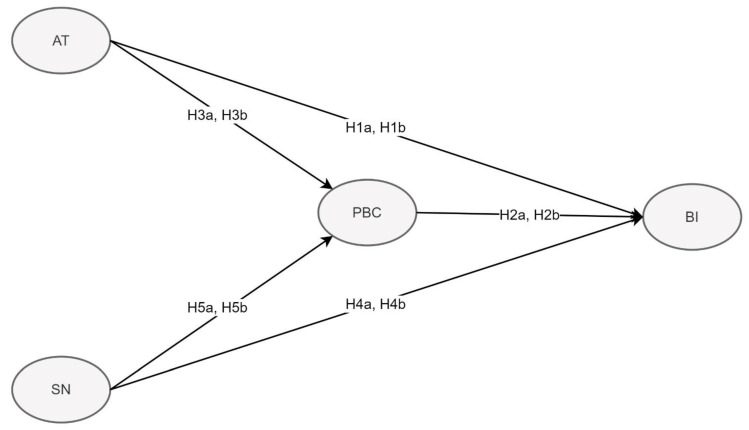
Study Hypotheses.

**Figure 3 ijerph-20-02518-f003:**
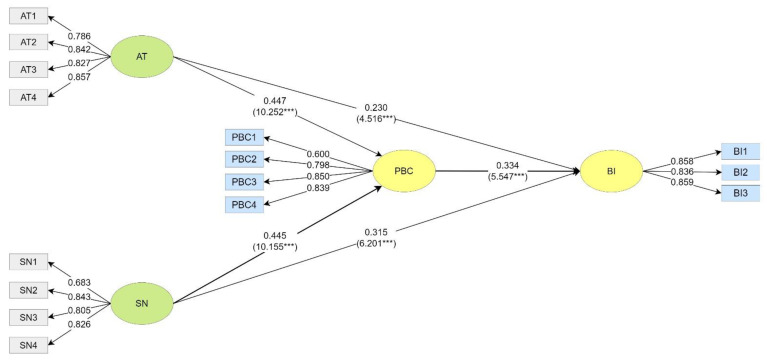
Path Coefficients of the Greater Taipei Metropolitan Group. *** *p* < 0.001.

**Figure 4 ijerph-20-02518-f004:**
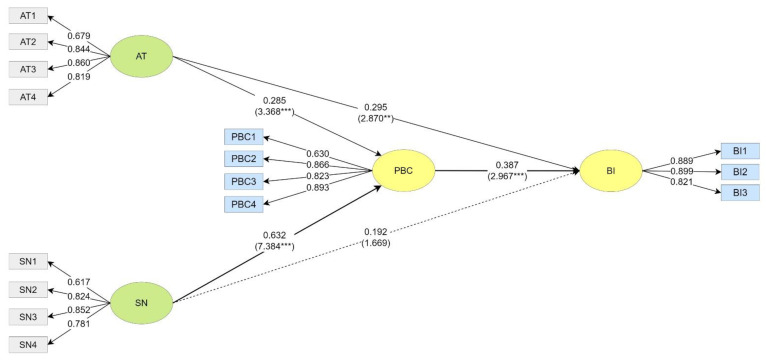
Path Coefficients of the New Taipei City Group. ** *p* < 0.01, *** *p* < 0.001.

**Table 1 ijerph-20-02518-t001:** Results of Attitude (AT) Question.

Attitude	Greater Taipei Metropolitan Group (*n* = 483)		New Taipei City Group (*n* = 127)	
	Mean	SD	Mean	SD
AT 1. I care about the air polluted with PM2.5 and effects on health	3.96	0.72	4.09	0.64
AT 2. I care about environmental issues arising from industry	4.11	0.66	4.24	0.70
AT 3. I care about environmental issues from economic development	4.04	0.70	4.14	0.66
AT 4. I care about whether the air polluted with PM2.5 affects the health of my family and I	4.17	0.73	4.27	0.65
AT Scores	4.07	0.58	4.19	0.53

**Table 2 ijerph-20-02518-t002:** Results of Subjective Norm (SN) Questions.

Subjective Norm	Greater Taipei Metropolitan Group (*n* = 483)		New Taipei City Group (*n* = 127)	
	Mean	SD	Mean	SD
SN 1. Most people who are important to me support me by not eating barbecued food	3.78	0.78	3.92	0.80
SN 2. Most people who are important to me support me by walking, cycling, or taking public transportation to go out	3.99	0.70	4.06	0.70
SN 3. Most people who are important to me support me when I wear masks for myself and my children when air pollution occurs	4.09	0.68	4.13	0.65
SN 4. Most people who are important to me support me when I participate in environmental protection activities to improve air pollution	4.00	0.68	4.13	0.66
SN Scores	3.97	0.56	4.06	0.54

**Table 3 ijerph-20-02518-t003:** Results of Perceived Behavioral Control (PBC) Questions.

Perceived Behavioral Control	Greater Taipei Metropolitan Group (*n* = 483)		New Taipei City Group (*n* = 127)	
	Mean	SD	Mean	SD
PBC 1. I can skip eating barbecued food to reduce air pollution	3.80	0.78	3.92	0.78
PBC 2. I can take public transportation, bicycle, or walk to reduce the problem of air pollution	3.99	0.72	4.05	0.81
PBC 3. I can remind my children to wear a mask when the air polluted with PM2.5 is serious even though wearing a mask is troublesome	4.15	0.72	4.23	0.66
PBC 4. I can guide children to wear a mask when the air polluted with PM2.5 is serious	4.18	0.66	4.25	0.67
PBC Scores	4.03	0.55	4.11	0.59

**Table 4 ijerph-20-02518-t004:** Results of Behavioral Intention (BI) Questions.

Behavioral Intention	Greater Taipei Metropolitan Group (*n* = 483)		New Taipei City Group (*n* = 127)	
	Mean	SD	Mean	SD
BI 1. I will let the child stay indoors when the Air Quality Index reaches a serious level	4.04	0.71	4.10	0.69
BI 2. I will pay attention to the Air Quality Index every day and remind children to pay attention to air pollution protection	3.98	0.74	4.02	0.72
BI 3. I will encourage the child to wear a mask when the air pollution is serious	4.15	0.67	4.18	0.67
BI Scores	4.06	0.60	4.09	0.60

**Table 5 ijerph-20-02518-t005:** Parents of the Greater Taipei Metropolitan Group.

	AT	SN	PBC	BI
AT	1.000			
SN	0.669 ***	1.000		
PBC	0.735 ***	0.755 ***	1.000	
BI	0.689 ***	0.703 ***	0.722 ***	1.000

*** = *p* < 0.001 Two-tailed test.

**Table 6 ijerph-20-02518-t006:** Parents of the New Taipei City Group.

	AT	SN	PBC	BI
AT	1.000			
SN	0.652 ***	1.000		
PBC	0.684 ***	0.821 ***	1.000	
BI	0.692 ***	0.678 ***	0.726 ***	1.000

*** = *p* < 0.001 Two-tailed test.

**Table 7 ijerph-20-02518-t007:** PLS Analysis of the Greater Taipei Metropolitan Group.

	AVE	CR	R^2^	Cronbach’s α
AT	0.686	0.897		0.848
SN	0.627	0.870		0.801
PBC	0.606	0.858	0.632	0.778
BI	0.724	0.887	0.667	0.810

**Table 8 ijerph-20-02518-t008:** PLS Analysis of the New Taipei City Group.

	AVE	CR	R^2^	Cronbach’s α
AT	0.646	0.879		0.814
SN	0.599	0.855		0.773
PBC	0.655	0.882	0.712	0.820
BI	0.757	0.903	0.619	0.839

**Table 9 ijerph-20-02518-t009:** Comparison of results: Greater Taipei Metropolitan Group and New Taipei City Group.

Greater Taipei Metropolitan Group				
	AVE	CR	R^2^	Cronbach’s α
AT	0.686	0.897		0.848
SN	0.627	0.870		0.801
PBC	0.606	0.858	0.632	0.778
BI	0.724	0.887	0.667	0.810
**New Taipei City Group**				
	**AVE**	**CR**	**R^2^**	**Cronbach’s α**
AT	0.646	0.879		0.814
SN	0.599	0.855		0.773
PBC	0.655	0.882	0.712	0.820
BI	0.757	0.903	0.619	0.839

**Table 10 ijerph-20-02518-t010:** *t*-test of Greater Taipei Metropolitan Group (GT) and New Taipei City Group (NT). (GT: *n* = 483, NT: *n* = 127).

	Mean	SD	T	P
AT (GT)	4.069	0.584	−2.141	0.033 *
AT (NT)	4.185	0.530		
SN (GT)	3.965	0.560	−1.816	0.071
SN (NT)	4.063	0.537		
PBC (GT) PBC (NT)	4.033 4.112	0.554 0.586	−1.424	0.155
BI (GT) BI (NT)	4.057	0.601 0.602	−0.708	0.480
	4.100			

* = *p* < 0.05 Two-tailed test.

## Data Availability

Data sharing is not applicable. All data generated or analyzed during this study are included in this published article.

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
