# Peer review of "Predicting the Protective Behavioral Intentions for Parents with Young Children Living in Taipei City and New Taipei City Using the Theory of Planned Behavior for Air Polluted with PM2.5"

_ijerph, 2023, doi:10.3390/ijerph20032518_

Round 1

Reviewer 1 Report

1.I'm glad to read this interesting article, which discusses the impact of air pollution on individual behavioral intentions. The issue of air pollution is closely related to sustainable development, and its discussion is of great value to the whole social environment. However, how air pollution affects human behavior at the micro level needs to be carefully discussed, especially the change of parental protection behavior at the family level needs to be rigorously verified by a more procedural process.

2.This paper draws on the theoretical framework of planned behavior to carry out the empirical analysis of questionnaire development, and it is necessary to fully sort out the application scenarios of the theory of planned behavior in the field of environmental pollution, especially in the field of air pollution.

3. Questionnaire development and design is a complicated work, which is also easy to fail. The rigor of the process needs a lot of literature support, and the standardization of the procedure needs to be reflected in the paper, such as the preliminary literature research, interview outline sorting, interview text data sorting and analysis, interview text saturation testing, necessary experiments and comparative control links, etc., all need to be more consistent with the research paradigm.

4. The overall structure of the article also needs to be further optimized, for example, the second part is not a necessary independent part, for example, the theoretical construction and model construction need to be independent. Before the questionnaire development and empirical study, a lot of space is needed to strongly support the elaboration of constructs.

5. There are also big problems in the introduction writing of the article. The logical level is chaotic, the semantics is complicated, and the target of the introduction is unclear. According to the research topic of the article, we can try to sort out more supportive literature to answer the following three questions: (1) What do you want to study? (2) Why do you want to study? (3) What will we learn? That leads to a better conversation with the reader.

6. The discussion of the article is an important part, and both theoretical and practical contributions need to be fully explained.

7. The language use of the article is also a big obstacle, and the beauty and standardization of the charts need to be strengthened.

Reviewer 2 Report

Introduction

Although this study aims to explore the effects of attitudes (AT), subjective norms (SN), and perceived behavioral controls (PBC) on the behavioral intentions (BI) of parents with young children, there are no statements or arguments about such relationships in the introduction. It is necessary to explain why researchers want to predict the behavioral intentions of parents. Researchers need to state their research justification, which should reflect the study hypotheses, in the introduction.

Theories and hypotheses

Researchers should provide a literature review to support the study hypotheses (H1a–H5b). They must explain the theory of planned behavior and the mechanism through which AT, SN, and PBC shape BI based on this theory. Also, the researchers need to define AT, SN, PBC, and BI, as well as provide empirical evidence of previous studies on the study hypotheses. 

Methodology

Researchers should provide additional explanations of the demographic characteristics of respondents, such as gender, education, occupation, and income levels. 

The researchers should conduct a confirmatory factor analysis to check the validity of the measurement.

Conclusions

The author(s) should strengthen the discussion of the study’s findings. The current discussion of the implications appears somewhat cursory. It is important to consider what we can do with the empirical evidence of this study. Finally, the author(s) should provide more specific managerial implications.

Round 2

Reviewer 1 Report

Great progress. Looking forward to its publication.

Reviewer 2 Report

Nice work, I am looking forward to seeing it in print.